# Thermal Activation of Coal Gangue with Low Al/Si Ratio as Supplementary Cementitious Materials

**DOI:** 10.3390/molecules27217268

**Published:** 2022-10-26

**Authors:** Xianli Yuan, Hong Wu, Ping Wang, Fen Xu, Shuang Ding

**Affiliations:** 1College of Environmental and Chemical Engineering, Dalian University, Dalian 116622, China; 2School of Chemical and Materials Engineering, Liupanshui Normal University, Liupanshui 553004, China; 3Guizhou Provincial Key Laboratory of Coal Clean Utilization, Liupanshui 553004, China

**Keywords:** coal gangue, low Al/Si ratio, thermal activation, supplementary cementitious material, physical properties, pozzolanic reactivity

## Abstract

To effectively utilize coal gangue (CG) with low Al/Si ratio, the thermal activation method was used. The activated CG, as supplementary cementitious materials (SCMs), was added into ordinary Portland cement (OPC) to study its physical properties. The XRD results show that CG undergoes a phase transition from kaolinite to metakaolinite during activation. The NMR tests reveal that the low polymerization state Q_3_ is continuously broadened, and the Al coordination gradually changes from Al ^VI^ to Al ^V^ and Al ^IV^. The CG particles are scale-like and glassy with a loose structure. By mixing the activated CG (under 800 °C) with cement (mass ratio = 3:7), the water demand of normal consistency increases by 7.2% and the initial and final setting times extend by 67 min and 81 min, respectively. The rough surface and loose structure of activated CG are the main factors contributing to the higher water demand of normal consistency. The micro-aggregate effect of the activated CG reduces the contact rate between the cement particles and water, and the interparticles, thus slowing down the process of hydration reaction, and leading to longer setting times.

## 1. Introduction

Coal gangue (CG), a solid waste from coal mining and processing, has low carbon content and high hardness [1,2]. Notably, the annual generation of CG solid waste reaches up to 37–550 million tons [3], and its effective utilization is still less than 15% [4]. Long-term accumulation of CG in the open air can lead to land occupation, spontaneous combustion, acid rain, underground infiltration, siltation of rivers, photochemical smog and mudslides, etc., that cause significant issues on environmental and human health [5,6,7].

The main chemical composition of CG is Al_2_O_3_ and SiO_2_, and CG can be classified into high Al/Si ratio CG and low Al/Si ratio CG [8]. Up to now, the CG with high Al/Si ratio is effectively applied on alumina production [9,10,11]. However, the CG with low Al/Si ratio hardly obtains large-scale application. Meanwhile, in most areas, the CG with low Al/Si ratio has a higher content than the CG with high Al/Si ratio, due to the geological conditions of mineralization and formation mechanisms. It should be mentioned that the CG with low Al/Si ratio has stable crystal structure, unstable composition, and low pozzolanic activity, thus limiting its high-value large-scale application [12]. Therefore, to enhance pozzolanic activity, the activation was adopted on CG with low Al/Si ratio.

In 2016, global production of blended and unblended cement reached 4.1 billion tons per year, causing environmental pollution [13]. Compared with the ordinary Portland cement (OPC) firing process, the CG calcination has a lower carbon footprint, and it will become one of the most promising supplementary cementitious materials (SCMs) [14]. Currently, SCMs include fly ash, finely ground blast furnace slag, silica fume, calcined clays, and natural pozzolans which are widely used, while CG is less utilized due to its lower activity. Thermal activation (calcination) is the most direct activation method, which is helpful to remove bound water and cations (such as Ca, Mg, and Fe) in CG. It also prevents the transformation of Si-O tetrahedra and Al-O octahedra from the fully polymerized state into long chain structure [15,16]. Via thermal activation, the major mineral phase composition of CG is quartz, kaolinite, and muscovite, which can release active SiO_2_ and Al_2_O_3_ [17,18]. In the literature, the optimal activation temperature of CG is 700 °C [19,20] or 800 °C [21,22].

CG as an additive in OPC has been proven to produce concrete with good mechanical properties [23,24,25]. Song et al. [26] found that when 20% activated CG is added, the 28-day strength of cementitious materials can reach 45.9 MPa, according with the standard of 42.5 grade cement. Zhao et al. [27] have shown that the use of CG as SCMs significantly accelerates the solidification and compressive strength development of the composite. Moreover, activated CG can also be incorporated into concrete as an aggregate to improve its mechanical strength [28,29,30]. Though the activated CG be broadly used as SCMs, less research was carried out to study its physical properties, which are essential factors in the research on cementitious materials.

Herein, the CG sample with low Al/Si ratio were activated and analyzed. Initially, the chemical and mineral composition was tested by XRD and XRF. Thermal method was used to activate the CG. Then, XRD, NMR, and FTIR tests were used to deeply study the characteristics of crystalline phase changes, alternating Al and Si coordination environments, and functional group changes during thermal activation. Moreover, the relationship between the specific surface area and microscopic morphological was analyzed. At last, the CG was mixed with cement to determine its water demand of normal consistency and setting time.

## 2. Materials and Methods

### 2.1. Materials

The CG samples were collected from the Ganjiagou Coal Mine (Guizhou, China). P.O.42.5 (Wumengshan, loss on ignition: 2.50) cement from Shuicheng Ruian Cement Plant (Guizhou, China) [31].

### 2.2. Thermal Activation of CG

The CG samples were crushed by a jaw crusher (Chuangweilai, PE100 × 60, Changsha, China) and ground under a 75 μm standard sieve. The thermal activation of the CG samples was carried out in a muffle furnace, which was heated from room temperature to 500 °C, 600 °C, 700 °C, 800 °C, or 900 °C. Each sample was held at these temperatures for two hours, then the samples were removed and quenched in air. The cooled samples were sealed and stored for later use. The preparation process of the thermally activated CG samples is depicted in Figure 1.

### 2.3. Characteristics of Activated CG

#### 2.3.1. Composition and Microstructure Analysis

The TG-DSC technique was used to analyze the thermal behavior of the CG samples. The samples were put into a simultaneous thermal instrument (Mettler Toledo, TGA-DSC1, Zurich, Switzerland) and were heated from room temperature to 1000 °C with a heating rate of 10 °C/min under N_2_ flow. The mineralogical composition of the samples was determined using an XRD instrument (Rigaku, Ultima IV, Rigaku Corporation, Japan) with Cu Kα radiation (l=1.54 Å, 40 kV, 40 mA). The scanning range 2*θ* was from 5° to 90° with a scanning speed of 4°/min and a step of 0.02°.

The chemical shifts of ^29^Si and ^27^Al NMR of the CG samples were analyzed by applying an NMR spectrometer (Bruker, AVANCE III HD 400, Karlsruhe Germany) and MAS experiment of ^27^Al and ^29^Si with a Solid Probe (4 mm/15 kHz). ^27^Al MAS NMR spectra were gathered at 104.198 MHz with a spinning speed of 12 kHz. The decomposition of ^27^Al and ^29^Si MAS NMR spectra was performed by fitting Gauss linear peaks employing the PEAKFIT simulation program based on the Gauss–Lorentz iteration method. The Lorenz–Gaussian ratio results were obtained with a squared correlation of r^2^ larger than 0.95. The infrared spectroscopy was performed under the diffuse reflectance condition using a Fourier transform infrared spectrometer. An infrared spectrometer (Thermo Scientific, NicoletiS5, Billerica, MA, USA) was applied (32 scans at 4 cm^−1^ resolution) within the range from 400 to 4000 cm^−1^. Before analysis, the CG powder was mixed with KBr and pressed into tablet form (2 mg sample in 200 mg kilned KBr).

#### 2.3.2. Hole Structure and Micromorphological Analysis

The micromorphologies of the calcined CG samples were observed by scanning electron microscopy (Zeiss, Sigma 500, Oberkochen Germany) and the whole structure and compositional analysis were carried out by an energy-dispersive spectrometry (BET-EDS). Additionally, the samples were plated with gold using a sputter coater to enhance the electrical permittivity before the SEM characterization. The BET specific surface was calculated from the nitrogen adsorption isotherms at −196 °C using an automatic specific surface analyzer (Micromeritics, TriStar II 3020 V1.01, Georgia, USA). The equilibrium interval was 5 s in the experiment. The adsorption and desorption isotherm curves were plotted according to the corresponding quantity adsorbed at relative pressures.

### 2.4. The Physical Property of Activated CG

The P.O.42.5 cement and thermally activated CG samples were used. The chemical composition of the three samples is shown in Table 1. The CG calcined at 500 °C is denoted as 500 °C-CG, the same as 800 °C-CG. The binary cementitious materials were fabricated by mixing the activated CG (mass ration: 30%) and pure cement (70%), according to the National Standard of the People’s Republic of China (GB/T1346-2001).

## 3. Results and Discussion

### 3.1. The Characteristics of the Raw and Calcined CG

#### 3.1.1. Mineral Phase Transition Analysis for the Thermal Decomposition

The XRD patterns of the CG sample under different calcination conditions are exhibited in Figure 2a. Kaolinite (Al_2_[Si_2_O_5_] (OH)_4_) was the major mineral phase in the clay minerals. As shown in Figure 3, kaolinite was a layered silicate mineral consisting of SiO_4_ tetrahedral planes connected by oxygen atoms parallel to the AlO_2_(OH)_4_ octahedral plane [32]. The kaolinite diffraction peaks of the activated samples were weaker than the raw sample. The kaolinite diffraction peak broadened and disappeared at around 800 °C which demonstrated that the kaolinite structure was disrupted by the removal of structural water at high temperatures, and the characteristic bands of amorphous phases was generated [33].

Furthermore, the crystal diffraction peaks were shifted and the planar spacing (d) was altered after the thermal activation as shown in Figure 2b. In comparison with the raw CG sample, the kaolinite diffraction peak of the calcined CG sample was shifted to a higher angle. This is due to that the disruption of the stable silicon (aluminum) oxygen tetrahedral structure in kaolinite under high temperature conditions. Meanwhile, the quartz diffraction peak was only slightly shifted to a higher angle at 900 °C, which might be due to the transition from the β-quartz to the β-scale quartz [34]. The d-values (~20.84 and ~26.62) of the two peaks of quartz in the raw CG sample were 4.259 Å and 3.3459 Å, respectively, and with reducing values after calcination. Simultaneously, the d-values of the diffraction peaks at different positions of kaolinite were also changed. It can be inferred from these phenomena that the thermal activation caused a small number of defects in the crystal and lattice distortion.

Throughout the calcination process, the characteristic peaks of quartz were dense. This might be due to the loss of kaolinite structural water to produce amorphous Al_2_O_3_ and SiO_2_ increasing the proportion of quartz [35]. Furthermore, the relative content of hematite in the un-calcined CG sample was low, while a large amount of hematite was observed in the calcined CG sample. This might be attributed to the calcination of pyrite in the air atmosphere to generate hematite and sulfur dioxide whose chemical equation can be expressed as 4FeS_2_ (pyrite) + 11O_2_ → 2Fe_2_O_3_ (hematite) + 8SO_2_.

The TG-DSC curve of the CG sample in the N_2_ atmosphere is presented in Figure 4. This figure shows that the weight loss was mainly attributed to the thermal decomposition of kaolinite, muscovite, and other minerals. The weight loss of the CG sample was slight from room temperature to 415.6 °C, caused by the liberation of the adsorbed water and the decomposition and burning of combustible substances. CG samples started to lose weight substantially in the temperature range of 415.6 to 671.4 °C. At 483.6 °C, the DSC curve has an endothermic peak, which was predominantly due to the removal of the hydroxyl groups on the alumina octahedral structure in kaolinite as water and formation of meta-kaolinite as shown in Equation (1) [36]. This effect corresponded to a mass loss of 32.76%, therefore, it can be inferred that the decomposition of the CG sample started at 415.6 °C and lasted until around 671.4 °C.
Al_2_O_3_.2SiO_2_.2H_2_O (Kaolinite) → Al_2_O_3_.2SiO_2_ (Metakaolin) + 2H_2_O (g)(1)

#### 3.1.2. Micromorphological and Compositional Analysis

The scanning electron microscopy (SEM) of the CG samples calcined at different temperatures is shown in Figure 5. The results of the compositional analysis are shown in Table 2. It can be noted that the predominant elements of the CG samples were Al, Si, and O with the Al/Si ratio (mol/mol) ranging from 0.40 to 0.50, which was characterized by low content of aluminum. The XRF analysis showed that the main chemical compositions of the raw CG sample were SiO_2_, Al_2_O_3_, Fe_2_O_3_, and CaO. The Al/Si ratio was determined by Equation (2) following the EDS test [37].

According to the SEM image shown in Figure 5b, the microstructure of the un-calcined CG sample was dense. Some small pieces adhered around large agglomerates and the edges of the agglomerate structures were relatively rounded. Moreover, in Figure 5c, the vitrification of clay minerals was observed in the CG sample, and the material transformed into a glass-like amorphous body (glassy state). This process occurred mainly during the conversion of kaolinite contained in the CG sample into metakaolin after heating [38]. However, the CG sample had a ‘scale-like’ lamellar structure at 800 °C. The edges and corners of the 800 °C CG sample became uneven, and the surface was chaotic.

The structural water was removed which destroyed its original crystal form causing a low degree of structural order. Moreover, the loose structure of the CG sample after the activation was due to the volatilization of some carbon particles and organic substances during the calcination and the expansion of the kaolinite structure after heating [39,40].
(2)$$O=\frac{S}{X\times A}\times M$$
where *O* is the oxide content (%), *S* is the single element content (%), *X* is the number of atoms, *A* is the atomic weight of the element, and *M* is the molecular mass of the oxide.

#### 3.1.3. Characterization by NMR

The ^29^Si MAS NMR spectra of the un-calcined CG sample and its calcined products at 600 °C, 700 °C, and 800 °C are shown in Figure 6. The calculated relative intensity of ^29^Si NMR with various Q_n_ forms in the calcined CG sample by way of fitting the ^29^Si spectral lines with the deconvolution technique is given in Table 3.

As shown in Figure 6a, the Si-O polyhedron of the un-calcined CG sample was in an unsymmetrical polymeric structure. The ^29^Si spectra contained a single sharp peak at −91.57 ppm and −110.48 ppm for the un-calcined CG sample which can be assigned to a Q_3_ (3Al) and a Q_4_ (0Al) Si environment, respectively [41,42]. In the NMR test analysis, five ^29^Si-NMR signals were used to represent the degree of aggregation of silica-oxygen tetrahedra which was simplified as Q_n_ structure. Here, *n* (=0 to 4) denotes the number of Si-O-Si bridging oxygens [43].

The chemical shift was changed, and the spectral peaks were broadened indicating that the Si-O polyhedral structure of the CG sample was significantly changed after the calcination. Studies suggested that the peak broadening was attributed to the reaction of protons with Fe [44]. At 700 °C, the proportion of the oligomeric Q_3_ structure reached 84.85% of the maximum, while the corresponding high polymerization state Q_4_ appeared to be the lowest value. It demonstrated the disruption of the ordered structure of the CG sample and the occurrence of the pozzolanic activity. Additionally, the formation of amorphous SiO_2_ was speculated. Moreover, the calculated relative intensity of ^29^Si NMR with the Q_n_ structure showed that the pozzolanic activity of the CG calcined at 700 °C was higher than at other temperatures.

Figure 7^27^Al NMR shows the un-calcined and calcined CG samples, and Table 4 summarizes the calculated relative intensity of ^27^Al NMR with various structures in the calcined CG sample by the fitting of the ^27^Al spectral lines using the deconvolution technique. The spectra were complex with a sharp signal center at 61.2 ppm in the un-calcined CG. This corresponded to 6-coordinated Al (Al ^Ⅵ^) and the spinning side bands could be seen on either side of the strongest line. Researchers found that the spinning side bands could be removed by changing the spinning frequency of the MAS rotor [45,46].

As shown in Figure 7, the ^27^Al spectra of the CG sample calcined at 600 °C showed a decrease in the peak intensity at 3.30 ppm. Two new peaks at chemical shifts of 21.24 ppm and 60.52 ppm appeared which belonged to the 5- and 4-coordinated Al, respectively. The signal at 60.52 ppm was a major one with the highest intensity and width. These alterations may be owing to the removal of structural water from kaolinite in the CG samples after calcination.

The splitting of the ^27^Al tetrahedral signals at different calcination temperatures was observed, which could be assigned to the presence of the crystallographically non-equivalent Al. Furthermore, compared with other calcination temperatures, the content of 6-coordinated Al at 800 °C decreased considerably and 4-coordinated became the main form. The ^27^A1 spectra of the CG sample at 800 °C showed that the presence of 5- and 4-coordinated Al was already evident in the poorly crystalline samples [47]. Therefore, it could be considered that the CG sample gradually transformed to low crystallinity during the activation process producing amorphous substances and improving the pozzolanic activity of the CG sample.

#### 3.1.4. Characterization by IR Spectroscopy

The internal structure of the CG sample was transformed at various calcination temperatures, so the characteristic peak types could be judged based on different wavenumbers. The FTIR spectra results are displayed in Figure 8.

As observed in Figure 8, two bands at 3694 cm^−1^ and 3620 cm^−1^ were attributed to the stretching vibrational band of the external hydroxyl (structural water) and the stretching mode of Al-O octahedral of kaolinite of the internal hydroxyl (interlayer water), respectively [48]. The two bands were visible in the un-calcined CG sample. However, when the temperature continued to rise, it weakened significantly and disappeared completely at 700 °C. The band that appeared at 912 cm^−1^ represented the Al-OH bending mode of the endo-hydroxyl group, and the absence of a peak at 912 cm^−1^ of the calcined CG sample was caused by the fracture of the Al-OH band. The breakage of the Al-OH band is attributed to the -OH in the silica (aluminum) oxygen tetrahedra of kaolinite becoming unstable at high temperatures, thus binding to ambient hydrogen and being stripped as structural water [49]. A further indication was that at 700 °C, the hydroxyl group has been removed with the crystal structure being destroyed.

The appearance of a band at 1024 cm^−1^ corresponded to the stretching vibration mode of Si-O-Si, and the intensity of the band at 1024 cm^−1^ became stronger because of the depolymerization of SiO_2_ tetrahedra during the thermal treatment [50]. In the low-frequency region, the faint bands at 798 cm^−1^ and 639 cm^−1^ were related to the different vibrational modes of the gibbsite-like sheet of the octahedral surrounding the Al^3+^ ions [51,52]. The band located at 535 cm^−1^ was attributed to the bending vibration of Si-O-Al. A decrease in this characteristic absorption band demonstrated that the bonding of Si-O-Al became weaker. In the NMR analysis, the changes in the Al ligands during the calcination were revealed which coincided with the above results. Moreover, it is noteworthy that a new band appeared at 561 cm^−1^ after the thermal treatment indicating the formation of metakaolin. The strongest band in the 700–800 °C domain was consistent with the observation of the XRD analysis.

#### 3.1.5. Specific Surface Area and Pore Structure

Based on the above analyses, to obtain an obvious comparison of the activation CG at different temperatures, the 800 °C and 500 °C are selected.

The isotherms of nitrogen adsorption and desorption registered for the calcined CG sample are presented in Figure 9. According to the IUPAC classification [53], the isotherms belong to type IV and the hysteresis loops symbolized the H3 type. The shape of the hysteresis loop was characteristic of the irregular mesopores. The H3 type had no saturated adsorption platform indicating that the pore structure was very irregular. This reflected pores including flat slit structures, cracks, and wedge structures [54]. Furthermore, adsorption at low relative N_2_ pressures (x < 0.1) proved the attendance of small volume micropores.

These curves showed the pore size distribution of the calcined CG sample as shown in Figure 10. The existence of micropores, mesopores, and macropores in the pore structure of the CG sample was confirmed. The activated CG pore structure was dominated by large pores with a MODE of 65.73 nm at 500 °C and 74.62 nm at 800 °C suggesting that the most developed pore size of the material at 800 °C was larger than that at 500 °C. The analysis of Table 5 clearly demonstrates that the higher the calcination temperature of CG, the smaller the specific surface area. At 800 °C, the total pore volume was reduced by 0.0009 cm^3^/g compared to 500 °C. Such results can be explained by the decomposition of organic matter that occurs mainly at 500 °C. However, the decomposition of organic matter and some minerals in CG at high temperatures creates a severe collapse of the thin walls between the internal pores and bridge, hence leading to a larger pore size and a lower specific surface area at 800 °C.

### 3.2. The Physical Performance of Cementitious Materials

Activated CG samples were mixed into the cement as SCMs for studying the water demand of normal consistency and setting time. The water demand of normal consistency and setting time of samples are listed in Table 6. Thermally activated CG was incorporated into cement as SCMs, and the following reactions occurred during the hydration process:Ca_3_SiO_5_ + 5.2H_2_O → 1.8 (CaO) SiO_2_ (H_2_O)_4.0_ (am.) + 1.2Ca (OH)_2_ (cryst.)(R1)
Ca_2_SiO_5_ + 4.2H_2_O → 1.8 (CaO) SiO_2_ (H_2_O)_4.0_ (am.) + 0.2Ca (OH)_2_ (cryst.)(R2)
yCa (OH)_2_ + Al_2_O_3_ + SiO_2_ + (n-y) H_2_O → yCaO·SiO_2_·Al_2_O_3_·nH_2_O (am.)(R3)

#### 3.2.1. Water Demand of Normal Consistency

Compared with the cement without CG blending, the water demand of normal consistency of the blended cement increased by about 7%. Obviously, as the calcination temperature raised, the water requirement for the normal consistency of binary mixture grew. When mixed with the CG sample calcined at 800 °C, the water demand of normal consistency was the maximum. 

The above results were due to the loose and porous structure of CG after thermal activation, which could adsorb water into the inner space of its structure. Additionally, from the SEM results, it can be noticed that the surface of the activated CG was rough, angular and scaly, leading to that more water was required to lubricate the particle surface [55,56]. Some phenomenon occurs, including internal pores, and thin wall collapse between the bridge during the calcination due to carbon particles, organic matter volatilization, and kaolinite structure dehydration.

Additionally, water demand of normal consistency was increased because slaked lime transforms ettringite when it reacts with anhydrite, active alumina, and water (Reaction (3)), and the pozzolanic reactivity both consume water [57].

#### 3.2.2. Setting Time

The initial setting and final setting of the cementitious material were defined as the initiation of the solidification and subsequent hardening, separately [58]. The values of setting time are consistent with GB175-2007. The binary mixture has a longer setting time than the cement paste. For the cement paste, the gap between the initial and final setting times was 26 min, while those of the binary mixture were 32 min at 500 °C and 40 min at 800 °C, respectively.

After thermal activation, the silicon (aluminum) oxygen tetrahedral stability structure of the kaolinite component was destroyed, when the CG particle was mingled with water, Si, and Al atoms on the surface hydroxylate due to adsorption of OH with the result in a negative charge on the surface [59,60]. In the binary mixture, since the Ca (OH)_2_ crystals produced by cement hydration (Reactions (1) and (2)) were not charged, therefore no electrical gravitational forces and rapid particle coalescence occur [55,61]. The pozzolanic reaction requires optimum activity under maintenance conditions of 7 to 28 days [62]. In our tests, the pozzolanic reactivity (Reaction (3)) was incomplete, leading to a longer setting time. The reaction schematic is plotted in Figure 11. In addition, the micro-aggregate effect of the activated CG reduces the contact rate between the cement particles and water, and the interparticles, thus slowing down the process of hydration reaction, and leading to longer setting times [26,63]. Meanwhile, the pore size of the 800 °C-CG is greater than that of 500 °C-CG (Figure 10). The increased pore size could adsorb a higher amount of water, which decreases the effective water involved in the hydration reaction, thus resulting in an increased setting time.

## 4. Conclusions

In conclusion, the activated CG was produced and used as SCMs into binary cementitious materials. The results have been obtained as below: (1) during the thermal activation of the CG with low Al/Si ratio, the mineral phase transition from kaolinite to metakaolinite produced reactive SiO_2_ and Al_2_O_3_. The low polymerization state Q_3_ was continuously broadened, and the Al coordination gradually changed from Al ^VI^ to Al ^V^ and Al ^IV^; (2) in CG activation, the volatilization of carbon particles and organic matter volatilization, as well as dehydration of kaolinite produced internal pores leading to structure collapse, resulting in loose structure and increased pore size; (3) the rough surface and porous structure of activated CG can increase the standard consistency of water consumption, due to the micro-aggregate effect, thereby extending the setting times of binary cementitious materials. 

## Figures and Tables

**Figure 1 molecules-27-07268-f001:**
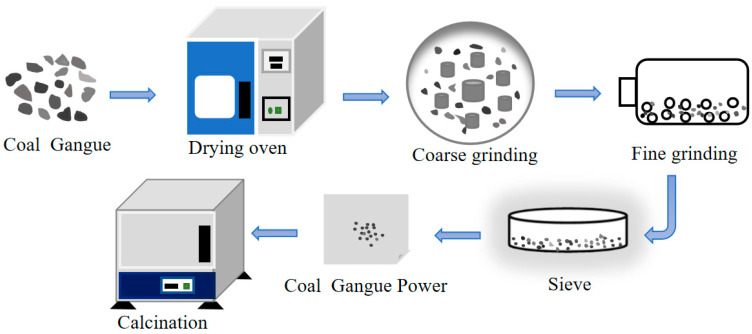
Preparation process of the thermally activated CG samples.

**Figure 2 molecules-27-07268-f002:**
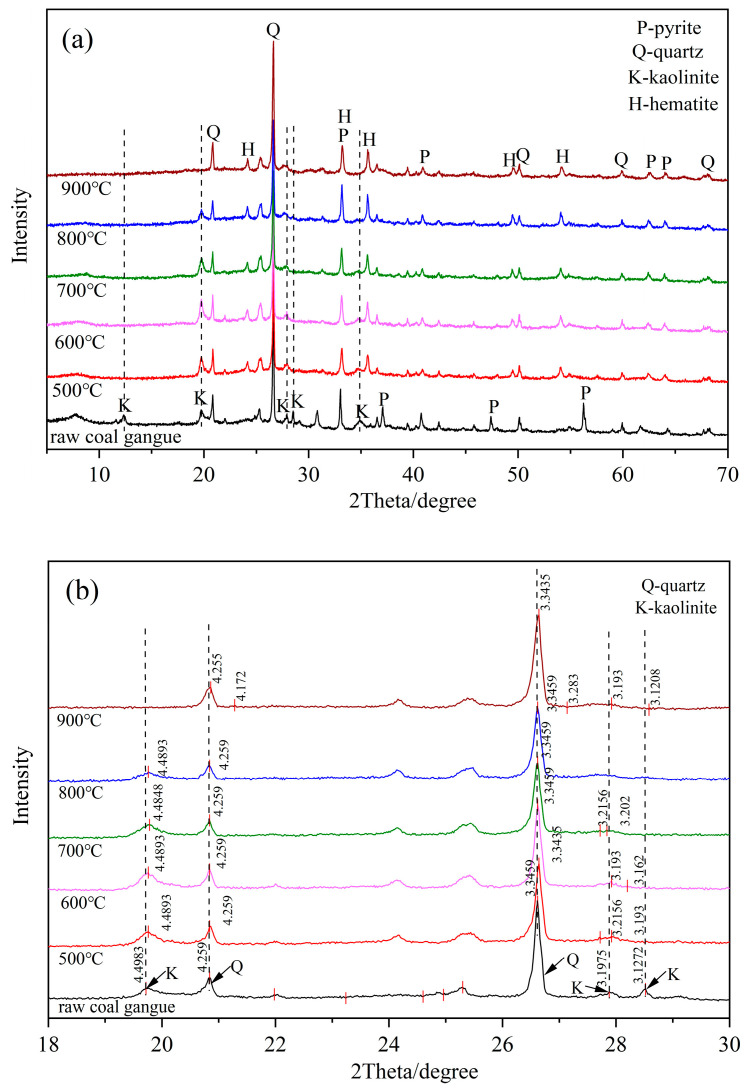
Mineralogy characteristics of the raw and calcined CG samples at different calcination temperatures (**a**) Crystal diffraction peak after thermal activation, (**b**) Plane spacing (d).

**Figure 3 molecules-27-07268-f003:**
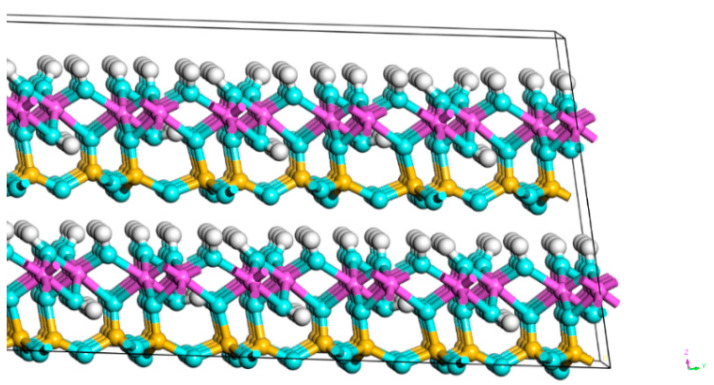
Ball-and-stick model of kaolinite crystal structure (white ball: H, blue ball: O, purple ball: Al, and yellow ball: Si).

**Figure 4 molecules-27-07268-f004:**
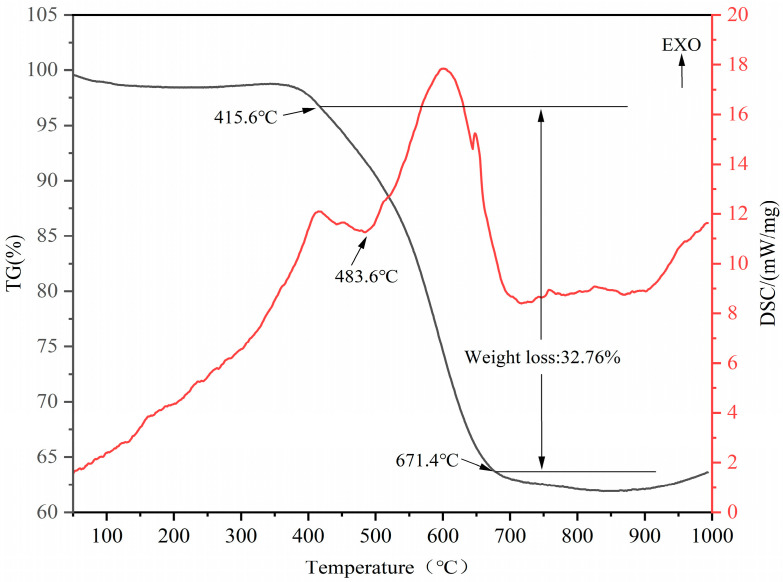
TG (left axis) and DSC (right axis) analysis of the raw CG sample under N_2_ flow.

**Figure 5 molecules-27-07268-f005:**
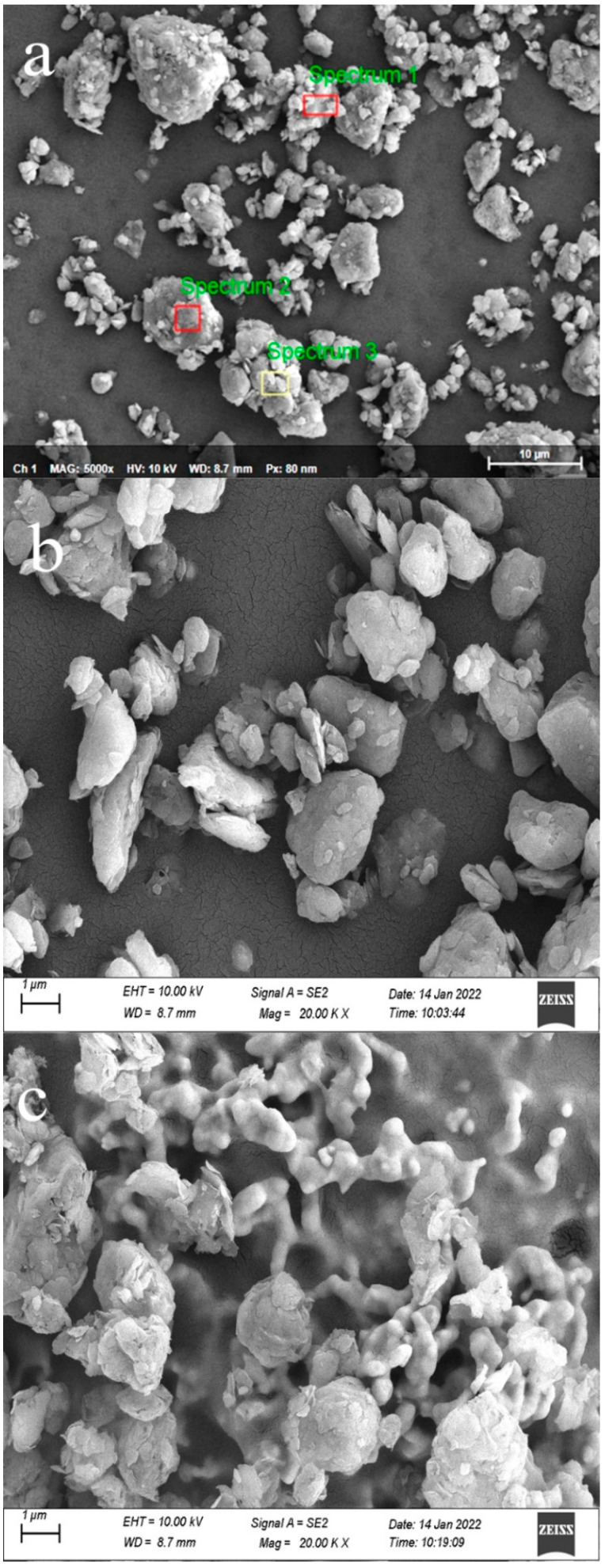
SEM images and EDS analysis of the CG sample (**a**) spot scanning, (**b**) un-calcined CG sample, and (**c**) calcined CG sample under 800 °C.

**Figure 6 molecules-27-07268-f006:**
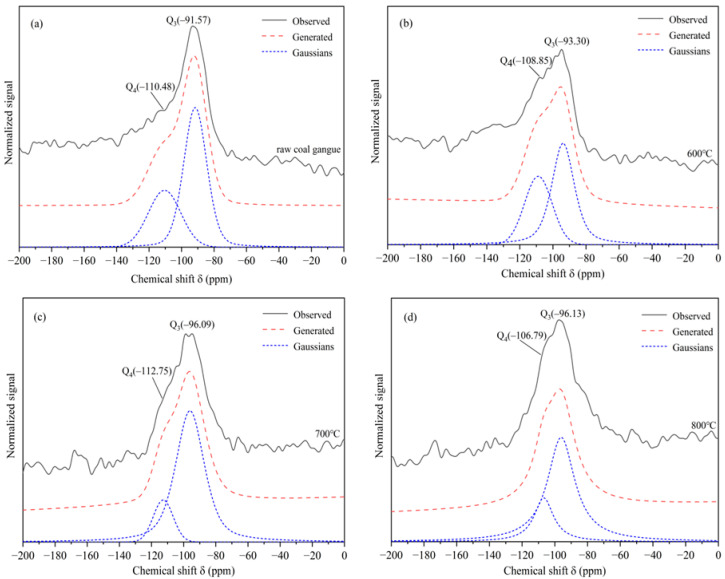
^29^Si NMR of the CG sample after calcination at different temperatures (**a**) un-calcined CG sample, (**b**) CG sample calcined at 600 °C, (**c**) CG sample calcined at 700 °C, and (**d**) CG sample calcined at 800 °C.

**Figure 7 molecules-27-07268-f007:**
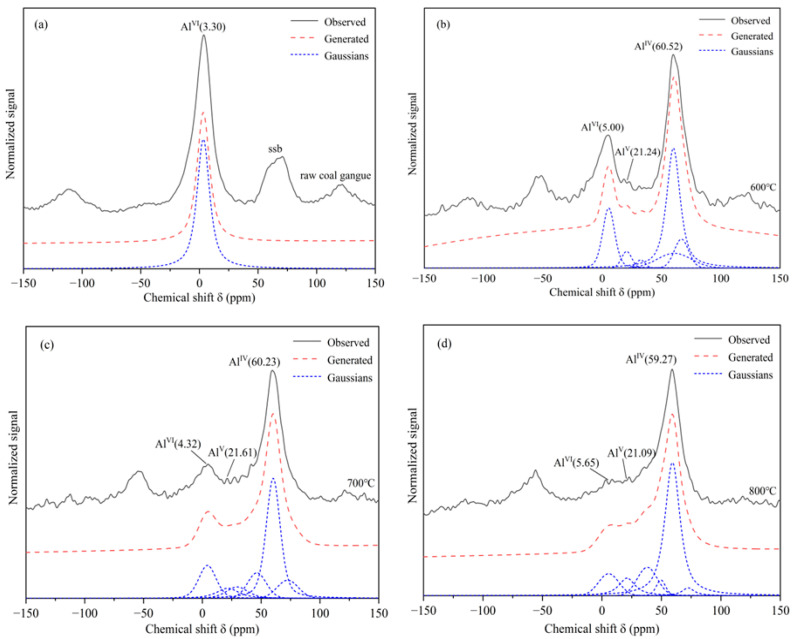
^27^Al NMR of the CG samples after the calcination at different temperatures (**a**) un-calcined CG sample, (**b**) CG sample calcined at 600 °C, (**c**) CG sample calcined at 700 °C, and (**d**) CG sample calcined at 800 °C.

**Figure 8 molecules-27-07268-f008:**
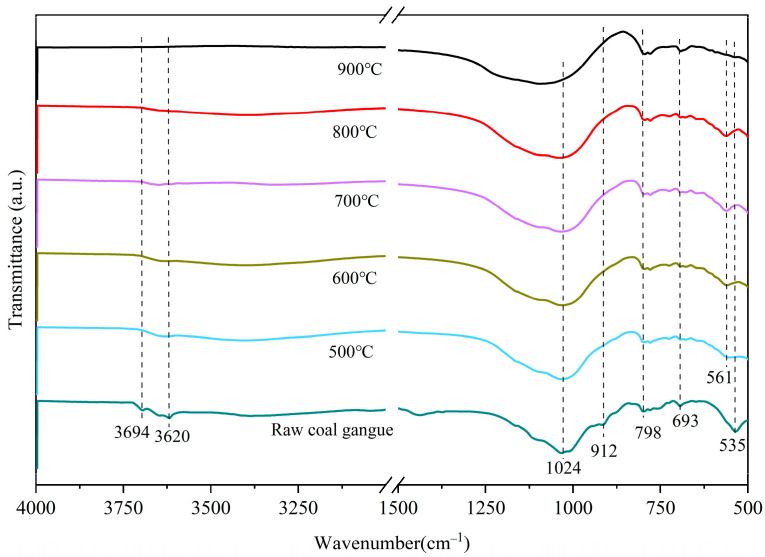
FTIR spectra of the CG sample calcined at different activation temperatures.

**Figure 9 molecules-27-07268-f009:**
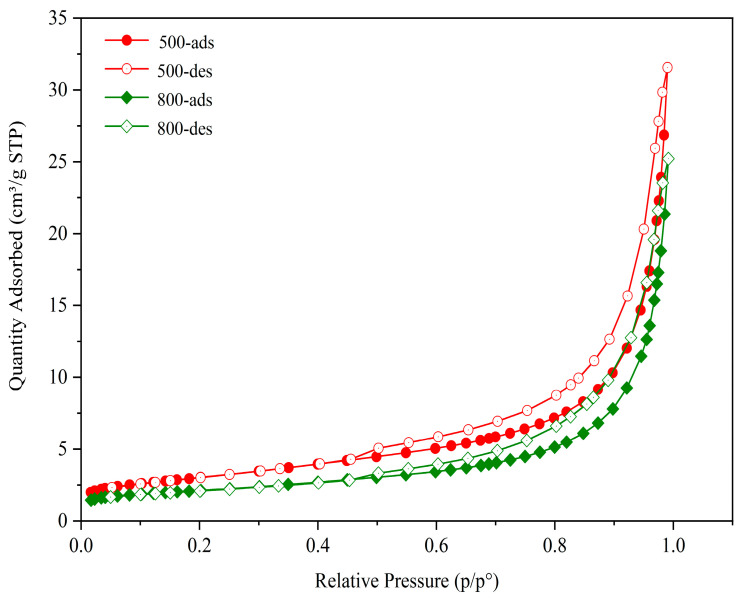
Isotherms of N_2_ adsorption and desorption on the CG-500 sample (calcined at 500 °C) and CG-800 sample (calcined at 800 °C).

**Figure 10 molecules-27-07268-f010:**
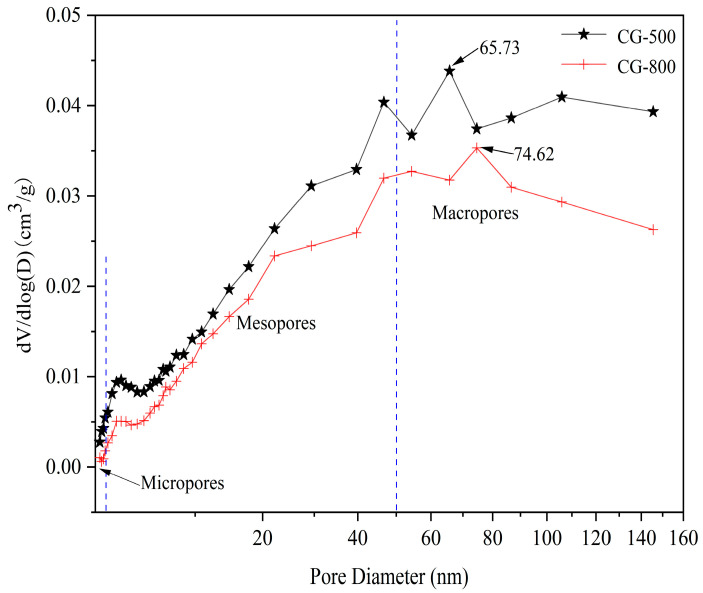
Pore size distribution concerning volume for the CG-500 sample (calcined at 500 °C) and CG-800 sample (calcined at 800 °C).

**Figure 11 molecules-27-07268-f011:**
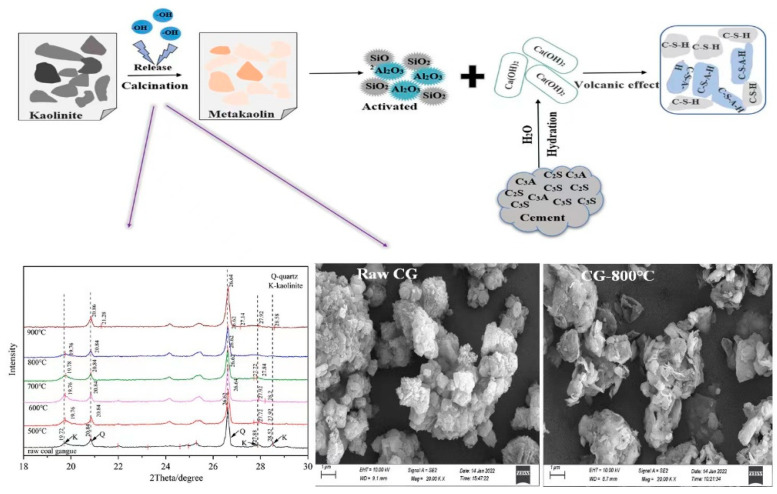
Schematic of the volcanic effect process of the active CG sample.

**Table 1 molecules-27-07268-t001:** The chemical composition of calcined CG and cement.

	SiO_2_	Al_2_O_3_	CaO	Fe_2_O_3_	SO_3_	MgO
500 °C-CG	43.4	10.3	2.57	31.5	0.16	/
800 °C-CG	40.5	10.6	3.61	32.2	0.15	0.21
Cement	/	/	/	/	2.79	3.52

**Table 2 molecules-27-07268-t002:** XRF and EDS elemental analysis of the raw CG sample.

Oxide	Content (wt %)	Elements		Content (wt %)	
	CG		Point-1	Point-2	Point-3
SiO_2_	35.90	O	48.01	48.85	45.89
Al_2_O_3_	12.70	Mg	3.19	0.91	1.19
CaO	4.06	Al	8.45	13.94	16.55
Fe_2_O_3_	26.40	Si	22.17	30.06	33.92
MgO	0.30	Ca	10.41	0.59	0.21
K_2_O	3.30	Fe	7.78	5.66	2.24
Al/Si(mol/mol)	0.42	/	0.40	0.49	0.50

**Table 3 molecules-27-07268-t003:** Calculated relative intensity of ^29^Si NMR with various Q_n_ structures in the CG samples.

Calcination Condition	Q_n_ Relative Intensity/%	
	Q_4_	Q_3_
Un-calcined CG	34.27	65.73
Calcined under 600 °C	39.76	60.23
Calcined under 700 °C	15.15	84.85
Calcined under 800 °C	22.98	77.02

**Table 4 molecules-27-07268-t004:** Calculated relative intensity of ^27^Al NMR with various structures in the calcined CG samples.

Calcination Condition	Al^3+^ Relative Intensity/%		
	Al ^Ⅵ^	Al ^Ⅴ^	Al ^Ⅳ^
Un-calcined CG	94.44	/	/
Calcined under 600 °C	12.85	6.11	34.30
Calcined under 700 °C	15.28	7.30	49.29
Calcined under 800 °C	9.77	9.11	60.09

**Table 5 molecules-27-07268-t005:** Specific surface area of the CG sample formed in the calcination process at different temperatures.

Calcination Temperature/°C	BET Surface Area/(m^2^/g)	Total Pore Volume/(cm^3^/g)
500	10.77	0.0031
800	7.54	0.0022

**Table 6 molecules-27-07268-t006:** The water demand of normal consistency and setting time of the CG sample calcined at different temperatures.

Calcination Temperature/°C	Water Demand of the Normal Consistency of Cement/%	Setting Time/Min	
		Initial	Final
Reference sample	28.80	261	287
500	35.44	318	350
800	36.00	328	368

Note: The reference sample was the cement sample without the addition of the CG sample because the raw CG sample did not have the pozzolanic activity and could not be used as SCMs.

## Data Availability

The data presented in this study are available in the article. This manuscript has not been published or submitted in whole or in part to any other journal. We believe that this manuscript will be of particular interest to the readers of your journal in a related research field. This manuscript has been thoroughly edited by a native English speaker from an editing company. An editing certificate will be provided upon request.

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
