# Peer review of "Thermal Activation of Coal Gangue with Low Al/Si Ratio as Supplementary Cementitious Materials"

_molecules, 2022, doi:10.3390/molecules27217268_

Round 1
Reviewer 1 Report
I accept the submitted article.
Author Response
We are very happy that the article was recognized, thank you very much for your hard work!
Reviewer 2 Report
The authors carried out research on the topic entitled "The manufacture of coal gangue additives with low Al/Si ratios and applied to binary cementitious materials". The impact of CG was studied at various exposure conditions. A representative fresh and microstructural investigations were carried out. In general, the article is well-written, well-structured, particularly the art work is excellent. The outcomes are also handy for the scientific community. However, there is a room for the improvement particularly in the methodology and some results. The specific comments can be found from the attached pdf file. After addressing the comments, it can be accepted.

Author Response
Thank you so much for all your hard work!

Reviewer 3 Report
The work is aimed to investigate the thermal activation of CG with low Al/Si ratio for use as additional cementitious materials. A positive aspect of the work is the use of a complex of physicochemical methods for studying the initial and activated CG samples. The disadvantage of the work is a weak study of binary mixtures of CG material with Portland cements.
There are the following remarks:
1. Abstract : The conditions for the preparation of CG, which was used in a binary cementitious material (composition not specified) and for which specific values are given for the characteristics of water demand and setting time, are not indicated here. It becomes clear from the full text that this refers to sample CG, thermally activated at 800°C. However, both of these characteristics are degraded compared to OPC and to a greater extent when 800°C is used, compared to 500°C. Therefore, it is necessary to correct and clearly write what benefits the use of activated CGs provides and what activation conditions are preferred.
2. Abstract: It is stated here that active Al2O3 reacts with gypsum to form ettringite. However, there is no gypsum in the composition of CG, which means that it comes from PC. As a part of PC, calcium aluminate reacts with gypsum and this happens very quickly. Therefore, it is not clear on what basis the conclusion is made about the formation of ettringite with the participation of active Al2O3.

Author Response
Thank you very much for your hard work!
